# Cautionary lessons from the COVID-19 pandemic: Healthcare systems grappled with the dual responsibility of delivering COVID-19 and non-COVID-19 care

Bhanu Duggal[1]*, Anuva Kapoor[2]*, Mona Duggal[3], Kangan Maria[4], Vasuki Rayapati[4], Mithlesh Chourase[4], Mukesh Kumar[4], Sujata Saunik[5], Praveen Gedam[6], Lakshminarayanan Subramanian[7]

1 Department of Cardiology, All India Institute of Medical Sciences (AIIMS), Rishikesh, Uttarakhand, India,
2 Community and Family Medicine Resident, All India Institute of Medical Sciences (AIIMS), Nagpur, India,
3 Advance Eye Centre, Post Graduate Institute of Medical Education and Research (PGIMER), Chandigarh, India, 4 Department of Cardiology, AIIMS, Rishikesh, Uttarakhand, India, 5 Health, Harvard T.H Chan School of Public Health, Boston, MA, United States of America, 6 National Health Authority, New Delhi, India, 7 Professor of Computer Science, New York University, New York, NY, United States of America

* bhanuduggal2@gmail.com (BD); anuvakapoor19@gmail.com (AK)

**Data Availability Statement:** The authors confirm that the minimal anonymized data set from our study is made available to be accessed via public

## Abstract

During the COVID-19 pandemic, hospitals were challenged to provide both COVID-19 and non-COVID treatment. **A survey questionnaire was designed and distributed via email to hospitals empanelled under the Ayushman Bharat–Pradhan Mantri Jan Arogya Yojana(AB-PMJAY), the world's largest National Health Insurance Scheme. Telephonic follow-ups were used to ensure participation in places with inadequate internet.** We applied support vector regression to quantify the hospital variables that affected the use vs. non-use of hospital services (Model-1), and factors impacting COVID-19 revenue and staffing levels (Model-2).We quantified the statistical significance of important input variables using Fisher's exact test. The survey, conducted early in the pandemic, included 461 hospitals across 20 states and union territories. Only 55.5% of hospitals were delivering emergency care, 26.7% were doing elective surgery and 36.7% providing obstetric services. Hospitals with adequate supplies of PPE, including N95 masks, and separate facilities designated for COVID-19 patients were more likely to continue providing emergency surgeries and services effectively. Data analysis revealed that large hospitals (> 250 beds) with adequate PPE and dedicated COVID-19 facilities continued both emergency and elective surgeries. Public hospitals were key in pandemic management, large private hospital systems were more likely to conduct non-COVID-19 surgeries, with not-for-profit hospitals performing slightly better. Public and large private not-for-profit hospitals faced fewer staff shortages and revenue declines. In contrast, smaller hospitals (< 50 beds) experienced significant staff attrition due to anxiety, stress and revenue losses. They requested government support for PPE supplies, staff training, testing kits, and special allowances for healthcare workers. The inclusion of COVID-19 coverage under AB-PMJAY improved access to healthcare for critical cases. Maintaining non-COVID-19 care during the pandemic indicates healthcare

data repository. The data file is uploaded as supporting information. For assistance in obtaining further details, please contact bhanuduggal2@gmail.com.

**Funding:** This research was funded by Health Technology Assessment (HTA), Department of Health Research (DHR), Ministry of Health & Family Welfare, Government of India (https://dhr.gov.in/health-technology-assessment-departmenthealth-research-dhr) under the grant number [F.no.t.11011/08/2017-HR(pART1)/8025571].

**Competing interests:** We have read the journal's policy and the authors of this manuscript have declared the following competing interests: Dr. Bhanu Duggal is a professor and Head of the Department at All India Institutes of Medical Sciences, Rishikesh, India. She was in the past also a professor and Head of Department at the Grant Medical College and Sir JJ group of hospitals. Also, she has worked with Escorts Heart Institute and Research Centre, New Delhi. She is a recipient of many awards, including the Diversity Travel Support Award in April 2013. Dr. Duggal is an internationally famed cardiologist with research in the utilization of stents in clinical practice in India. She also completed a fellowship from ICMR International Fellowship: Cleveland Clinic, USA Wellcome – DBT, and Early Career Clinical Fellow (Indo-UK Fellowship).

system resiliency. A state-wide data-driven system for ventilators, beds, and funding support for smaller hospitals, would improve patient care access and collaboration.

## 1. Introduction

COVID-19 hit different countries with varying intensity after it first appeared in Wuhan, China. It swept across the globe, humbling healthcare systems worldwide, and exposing their under-preparedness and weak response. During times of significant SARS-CoV-2 transmission, emergency care services were especially affected, directly impacting those who needed urgent medical care. 36% of the world's nations reported interruptions in ambulance facilities, 32% to 24-hour emergency services, and 23% to emergency surgery systems [1]. In addition, regular and nonemergency medical care access was also disrupted. Different hospital systems responded differently to the pandemic onslaught. Studies have been done to correlate hospital characteristics to COVID-19 outcomes [2]. One study from Boston, Massachusetts, found that critical COVID-19 patients admitted in hospitals with < 50 ICU beds had a higher risk of death [3, 4]. Supply chain disruptions of personal protective equipment (PPE) (e.g., disposable gloves, face shields, eye protection, isolation gowns, caps, surgical masks, N95 respirators), led to a shortage of these lifesaving medical supplies. Of utmost importance, was the effect on the mental and physical health of HCWs. They battled on the pandemic frontlines, underprepared, overburdened and overwhelmed as they contended with the shortage of PPE, inadequate preparation and absence of guidelines for handling this new highly infectious virion. The pressing need to isolate for fear of spreading the disease to family and friends triggered feelings of extreme loneliness [5]. Hospitals grappled with severe staff shortages, many times in critical care units. The health insurance policy of a country was also important in determining access to healthcare facilities, as many people were laid off due to national lockdowns. This survey was conducted to analyse the above challenges and responses of healthcare providers in India, across states and different healthcare systems to better prepare for future pandemics.

## 2. Materials and methods

This cross-sectional study included hospitals empanelled under Ayushman Bharat Yojana across 18 states and 2 Union Territories of India. All the participating hospitals were among the 20,347 hospitals registered on the National Health Authority (NHA) portal from which a state-wise list of registered email addresses was collected. An online webpage and app-based survey questionnaire [5] with multiple sections was developed and circulated to the 20,347 hospitals during May–July 2020. Of the 20,347 empanelled hospitals, considering the constraints and challenges of dealing with the start of the pandemic and to have representation from all states we fixed the 'target sample size' for responses as 5% of all hospitals from each state. Voluntary response sampling was adopted and non-responders were excluded. No specific allocation and selection of public and private hospitals was done. The survey questionnaire was divided into various sections; the initial section was to capture sociodemographic and infrastructure-related information about the hospitals. The latter sections consisted of challenges being faced by hospitals; in terms of supplies of protective equipment, testing, workforce shortage, decline in hospital services and revenue, measures/processes being taken by hospitals for continuation/resumption of hospital services and expected support from the government (AB-PMJAY). The original questionnaire was developed in English and was also translated into Hindi to improve the convenience of answering. The dual-language questionnaire was adopted into a fillable online version using custom software and was rendered in both- a web version and a smartphone version. After consenting to participate in the

survey, the webpage redirected the hospital administrator to the first section of the survey. The person responsible for filling up the questionnaire could be one among the managing director/ chief executive officer/ head of the hospital, or nodal officer appointed by the hospital for communication with the NHA. Respondents from each hospital were requested to fill out the questionnaire in the online mode using the link shared with them in the email. A smartphone version of the questionnaire was also developed and provided to the hospitals with an option for filling out responses offline and sending the filled questionnaire subsequently. After one week of sending the original email, a follow-up email was sent to the hospitals that did not respond to the first email owing to poor connectivity. Apart from this, to have a proportionate response rate from all states the survey team telephonically followed up with the hospitals that were facing issues related to poor connectivity. As the survey constituted closed-ended, multiple-option questions there was no scope of response bias or need for probing questions. All the responses were tabulated in the Excel spreadsheet, and analysis of descriptive statistics was done using IBM SPSS Statistics version 24.0. Data analysis was performed to understand the effect of hospital beds and types of hospitals on the continuity of hospital services including emergency services, obstetric services, OPD services, and elective surgery. The dependent variables were categorized as 1 if services were provided and 0 otherwise. We also generated heat maps to understand hospital variables, which could have affected hospital services (Model 1), and COVID-19-related revenue and staff shortages (Model 2). In the first model, 26 variables were considered and in the second model, 19 variables were considered. For Model 1, we considered 19 input variables and 7 output variables. Fig 1 shows a heat map of the cross-correlation across a broader array of input and output variables used in the survey that allowed us to choose specific input variables that correspond to the seven chosen output variables of interest relating to this model. We outline these variables below in more detail.

The survey collected information on the following input variables.

'has Screening For COVID-19': This variable refers to whether the hospital has a screening process in place to identify patients with COVID-19 symptoms. If the hospital has a screening process in place, it may be better equipped to handle COVID-19 cases and prevent the spread of the virus within the hospital.

'has Separate Facility For COVID-19': This variable refers to whether the hospital has a separate facility to treat COVID-19 patients. Having a separate facility can help prevent the spread of the virus within the hospital and ensure that COVID-19 patients receive the care they need.

'facing Management Challenges': This variable refers to whether the hospital is facing any management challenges related to COVID-19, such as shortages of staff, equipment, or supplies. If the hospital is facing challenges, it may have difficulty providing services to its patients.

'has Negative Pressure Room': This variable refers to whether the hospital has negative pressure rooms, which can help prevent the spread of infectious diseases by containing airborne particles. Having negative pressure rooms can be particularly important for hospitals treating COVID-19 patients.

'know that Ayushman Bharat Yojana' insurance and was: This variable refers to whether the hospital staff know the coverage of testing and treatment of COVID-19 under the Ayushman Bharat Yojana insurance scheme.

'Risk assessment for staff': This variable refers to whether the hospital has conducted a risk assessment for its staff. Knowing the risks can help the hospital take steps to protect its staff and prevent the spread of the virus within the hospital.

'adequate supply of Essentials' protective equipment: This variable refers to whether the hospital has adequate essential supplies in critical areas, such as PPE, N95 masks medications, and medical equipment. If the hospital does not have adequate supplies, it may have difficulty providing services to its patients.

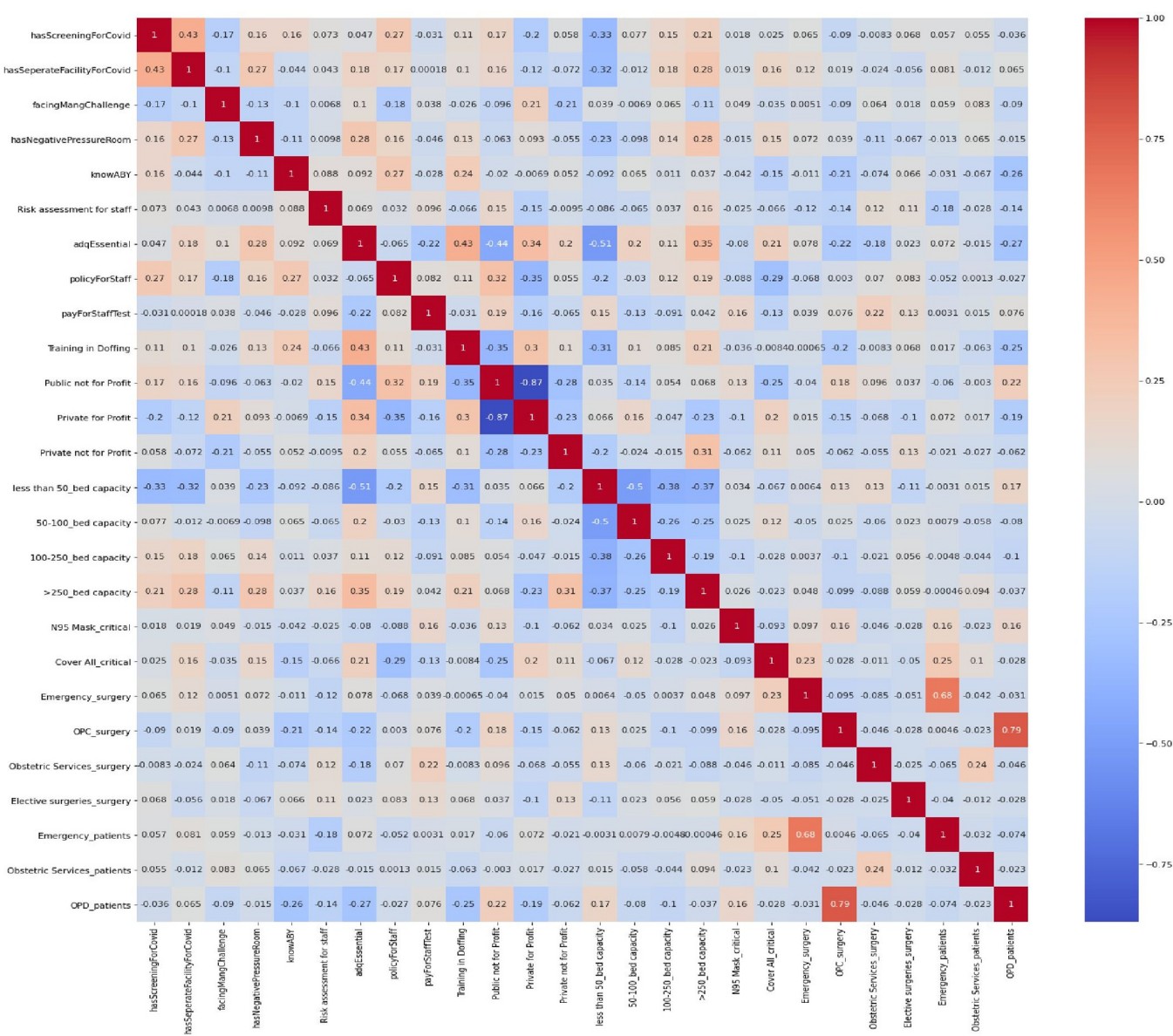

**Fig 1. Model 1 represents the relationship between various factors related to COVID screening and healthcare facility capabilities.**

'policyForStaff': This variable refers to whether the hospital has policies in place to protect its staff, such as guidelines for the use of PPE and procedures for dealing with COVID-19 cases. Having policies in place can help the hospital prevent the spread of the virus within the hospital and ensure that its staff are protected.

'payForStaffTest': This variable refers to whether the hospital is providing COVID-19 testing for its staff. If the hospital is providing testing, it can help prevent the spread of the virus within the hospital staff and ensure that its staff are protected.

'Training in Doffing': This variable refers to whether the hospital has provided training to its staff on how to safely remove PPE. Proper doffing procedures can help prevent the spread of the virus within the hospital.

'Public not for Profit', 'Private for Profit', and 'Private not for Profit': These variables indicate the ownership status of healthcare facilities. Understanding the ownership status is

important to identify the resources available for managing COVID-19 and to assess the role of the public and private sectors in the pandemic response.

'less than 50_bed capacity', '50–100_bed capacity', '100–250_bed capacity', '>250_bed capacity': These variables indicate the size of healthcare facilities, which is important in understanding the capacity to manage COVID-19 patients. Larger facilities may have more resources and specialized equipment to manage severe cases of COVID-19.

'N95 Mask_Critical: This variable represents the availability of N95 masks, which are critical for protecting HCWs and patients from airborne infectious diseases.

'Cover All_critical': This variable represents the availability of Coveralls, which are protective clothing worn by HCWs to prevent the spread of infectious diseases.

The output variable 'type of other Patients' refers to the type of patients other than COVID-19 patients, and it includes three categories: Emergency patients, Obstetric Services patients, and OPD patients. During the COVID-19 pandemic, the number of patients visiting hospitals for non-COVID-19-related health issues decreased due to various reasons, such as fear of contracting the virus, restrictions on travel and movement, and healthcare system overload. The variable 'type of Surgeries' refers to the type of surgeries, and it includes four categories: Emergency surgeries, Obstetric surgeries, OPD, and Elective surgeries.

For Model 2 (Fig 2), we considered 14 input variables and 7 output variables:

'Seven INSIGHT': This variable refers to the perception of increased stress, anxiety or fear in the daily life of healthcare providers during the COVID-19 pandemic.

'teleconsultation': This variable indicates the use of teleconsultation or telemedicine services to provide healthcare services remotely.

'Moderate training', 'Substantial training', 'Minimal training', 'No training': These variables refer to the level of training provided to HCWs in managing COVID-19.

Other input variables defined in model 1 that were considered were: organization type 'Public not for Profit', 'Private for Profit', 'Private not for Profit',

Hospital size is assessed by bed capacity. 'less than 50_bed capacity', '50–100_bed capacity', '100–250_bed capacity', '>250_bed capacity', 'has separate Facility For COVID-19'.

The output variable 'Reduction in staff' refers to the decrease in the number of HCWs employed by an organization, which could be due to various reasons such as decreased patient load or staff being infected with COVID-19. %" "reduction in revenue' reflects the financial impact of the pandemic on healthcare organizations, as they face decreased revenue due to reduced patient load or increased costs for PPE and other supplies. 'percentage of reduction ' variable further breaks down the reduction in revenue into different categories, such as '<10% and 10–19%', '20–29% and 30–39%', and '40–49% and >50%', providing a more detailed understanding of the financial impact on the organizations. When machine learning algorithms are applied, considerable performance problems in the predictions can be attributed to low data quality, especially in health data. Given this background, basic data pre-processing was applied to increase the performance of the models. It is a critical step in the data analysis process, as the quality of the data and the accuracy of the results depend on the quality of the pre-processing. Missing values in the input and output variables were dropped. One-hot encoding was performed on the categorical input variables. This allowed for the conversion of the variables into a numerical format suitable for statistical analysis. In one-hot encoding, each unique category or value in a categorical variable is represented by a binary feature. For example, suppose we have a categorical variable "type of surgery. One-hot encoding involves creating a new binary variable for each unique category in the categorical variable. In this case, we would create three new binary variables, one for each possible value in "type of surgery": "type of surgery_ Elective": 1 if the surgery is "Elective", 0 otherwise "type of surgery _Emergency": 1 if the surgery is "Emergency", 0 otherwise "type of surgery_OPD": 1 if the surgery is "OPD", 0 otherwise.

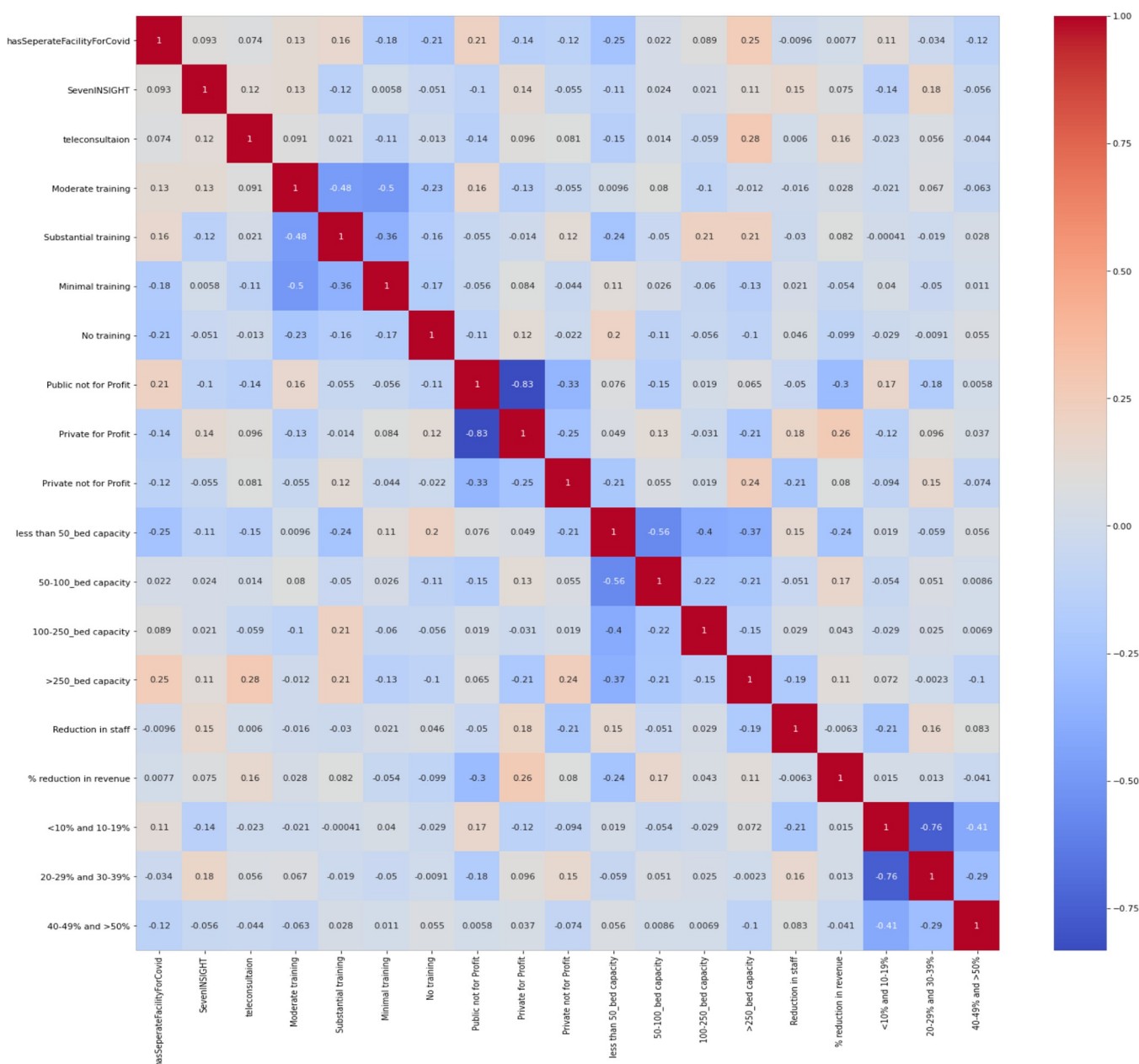

**Fig 2. Model 2 variables the rows and columns represent different variables, and each cell represents the correlation coefficient between two variables.**

Similarly, we performed this technique on other variables.

Fisher's exact test was used to evaluate the association between each input variable and the output variable in a dataset. Fisher's exact test is a statistical test used to determine if there is a significant association between two categorical variables. By performing Fisher's exact test, we determined the p-values for input variables that are significantly associated with the output variable, which can then be used to build a machine-learning model.

The matrix shows that a positive correlation between two variables indicates that an increase in one variable is associated with an increase in the other variable, while a negative correlation indicates that an increase in one variable is associated with a decrease in the other

variable. The strength of the correlation is indicated by the absolute value of the correlation coefficient, with larger absolute values indicating a stronger correlation.

In addition, bar plots were generated to show the number of hospitals in each category ("Yes" or "No") for the significant categorical variables. Support vector regression was utilized to predict the values of the output variables based on the input variables. The root mean squared error (RMSE) score was calculated to evaluate the performance of the model. We divide the dataset into 93% of the total samples for training, and the rest 7% for testing.

## 3. Results

### 3.1. State-wise distribution of the responses

We achieved a minimum 5% response rate in the majority of the states. A total of 461 hospitals from 18 states and 2 UTs of India responded to the survey. Initially, a higher response rate was found from well-performing states and those hospitals with better connectivity as per e- Readiness Index [7] shows the state-wise distribution of the proportion of these responses. Maximum responses were received from Tamil Nadu (12.4%), followed by Jharkhand (10.2%), Uttar Pradesh (10.2%), Bihar (9.3%), and Maharashtra (8.7%). 55.8% (n-257) hospitals that participated in the survey were public not-for-profit, 34.7% (n = 160) were private for-profit hospitals. and 8.7% were private not-for-profit hospitals (Table 1). 75.9% (n = 350) were non-teaching type followed by teaching at 11.1% (n = 51), others (minor teaching hospitals) at 8.9% (n = 41) and central institutes were 1.5% (n = 7).57.3% of hospitals (n = 264)had <50 beds 5; 21.7% (n = 100) had 50–100 beds, 10.2% (n = 47) had 100–250 beds and only 8% (n = 37) had >250 beds.

**Table 1. Characteristics of responding hospitals.**

| Hospital characteristics | No of Hospitals (N) | % |
|---|---|---|
| Nature of practice | | |
| Public not-for-profit | 257 | 55.8 |
| Private for-profit | 160 | 34.7 |
| Private not-for-profit | 40 | 8.7 |
| No response | 4 | 0.9 |
| Hospital type | | |
| Non-teaching | 350 | 75.9 |
| Teaching | 51 | 11.1 |
| Other | 41 | 8.9 |
| No response | 12 | 2.6 |
| Central Institute | 7 | 1.5 |
| Bed capacity | | |
| <50 | 264 | 57.3 |
| 50–100 | 100 | 21.7 |
| 100–250 | 47 | 10.2 |
| >250 | 37 | 8.0 |
| No response | 13 | 2.8 |
| Digital patient health record system | | |
| Yes | 217 | 47.1 |
| No | 187 | 40.6 |
| In process | 36 | 7.8 |
| No response | 21 | 4.6 |
| **Total** | **461** | **100.0** |

 

### 3.3. Changes in hospital services and availability of healthcare workers (Table 2)

73.3% of hospitals were providing outpatient consultations (including telemedicine consultations), 66.4% (n = 306) emergency services, and 43.8% (n = 202) obstetric services 43.8%. Of surgical procedures, emergency surgeries were being performed in 55.5% (n = 256) of hospitals only, obstetric in 36.7% (n = 169), elective 22.6% (n = 104), and 16.9% (n = 78) other surgeries (minor procedures etc.). Aerosol-generating procedures were being done by 14.5% (n = 67) hospitals only. A decrease in the number of OPD patients visiting the hospitals was reported in 55.1% (n = 254) of hospitals while a significant portion of the hospitals (44.9%) did not respond to this particular question. It is noteworthy that 50.5% of survey respondents felt that their hospitals had less than required HCWs of more than one category (doctors/ nurses/ administrative staff), and 49.5% of the respondents were experiencing a scarcity of staff in critical non-COVID-19 areas such as intensive care units and dialysis units.

**Table 2. Changes in hospital services and availability of manpower.**

|  | Yes (%) | No (%) | No response |
|---|---|---|---|
| Consultation of non-COVID-19 patients *Emergency* | *306 (66.4)* | *155 (33.6)* | |
| *Obstetric services* | *202 (43.8)* | *259 (56.2)* | |
| *OPD* | *338 (73.3)* | *123 (26.7)* | |
| *Elective (general) surgeries* | *109 (23.6)* | *352 (76.4)* | |
| *Others* | *116 (25.2)* | *345 (74.8)* | |
| Performing surgeries | | | |
| *Emergency* | *256 (55.5)* | *205 (44.5)* | |
| *Obstetric services* | *169 (36.7)* | *292 (63.3)* | |
| *Elective surgeries* | *104 (22.6)* | *357 (77.4)* | |
| *Others* | *78 (16.9)* | *383 (83.1)* | |
| *Aerosol generating procedure* | *67 (14.5)* | *329 (71.4)* | *65 (14.1)* |
| ***Change in OPD patients*** | | | |
| *less than 10%* | *81 (17.6)* | | |
| *10–19%* | *47 (10.2)* | | |
| *20–29%* | *53 (11.5)* | | |
| *30–39%* | *42 (9.1)* | | |
| *40–49%* | *15 (3.3)* | | |
| *>50%* | *16 (3.5)* | | |
| *Non-response* | *207 (44.9)* | | |
| *Total hospitals with change in OPD 254 (55.1)* | | | |
| Unavailability of staff | *200 (100)* | | |
| *Administrative staff* | *11 (5.5)* | | |
| *Nurses* | *21 (10.5)* | | |
| *Doctors* | *8 (4.0)* | | |
| *More than one type of staff* | *101 (50.5)* | | |
| *No response* | *59 (29.5)* | | |
| Adequate staff available for dialysis, *168 (36.4%)* cardiac and ICU services | | *228 (49.5)* | *65 (14.1)* |

### 3.4 Stress among HCWs and suggestions to improve mental health

Over sixty (61.2) % of healthcare providers who responded to the question perceived increased stress, anxiety, or fear while treating patients during COVID-19. Only 35% of the respondents expressed confidence in handling the pandemic. Most HCWs stressed the need to have sufficient PPEs and other measures to ameliorate anxiety, including increased testing and diagnostic facilities. HCWs also requested suitable quarantine facilities, fixed duty hours and financial security like COVID-19 insurance, in case of any pandemic morbidity or mortality. (Table 3)

### 3.5. Additional needs of HCWs

Inputs were provided by 44.9% of responders on their additional requirements. These suggestions can be categorized into five major categories: safety- and infrastructure-related (65.7%), administration-related

(39.6%), HCW care-related (14.5%), patients' care- and public-related (14%), and others (1%). Of all responders (136) who felt safety- and infrastructure-related needs majority (19.8%) felt that the availability of medicines, PPEs, masks, sanitisers etc. should be ensured

**Table 3. Stress among HCWs and suggestions to reduce it.**

| *How prepared do you feel to handle the COVID-19 pandemic?* | |
| --- | --- |
| Fully prepared/well prepared | 162 (35.1%) |
| Not prepared/least prepared | 34 (7.4%) |
| Moderately prepared/partially prepared | 30 (6.5%) |
| Prepared as per Govt guidelines | 16 (3.5%) |
| No comment/do not know/not sure | 160 (34.7%) |
| *As a healthcare provider do you feel any perceived stress/anxiety/fear in your routine during the COVID-19 pandemic* | |
| Yes | 282 (61.2%) |
| No | 111(24.1%) |
| No response | 68(14.7%) |
| *If yes, what are your suggestions to help reduce the stress among HCWs?* | |
| Protection related suggestions | 108 (23.4%) |
| PPE kits, masks and other protective measures | 64 (22.7%) |
| Yoga, pranayama, and other exercises for stress reduction | 28 (9.9%) |
| Regular screening of patients | 6 (2.1%) |
| Increase COVID-19 testing | 5 (1.8%) |
| Periodic staff screening | 3 (1.1%) |
| After performing surgeries treating doctors should be tested for COVID-19 | 1 (0.4%) |
| Postpone non-essential activities | 1 (0.4%) |
| Preparation, equipment and infrastructure-related suggestions | 72 (25.3%) |
| Training for awareness of COVID-19 | 35 (12.4%) |
| Clear guidelines from Govt/transparent information | 17 (6.0%) |
| Infrastructure development | 7 (2.5%) |
| Better knowledge about COVID-19 | 6 (2.1%) |
| Proper quarantine facilities | 5 (1.8%) |
| Separate hospitals for COVID-19 | 1 (0.4%) |
| Early discharge of patients | 1 (0.4%) |
| *Administrative support related suggestions* | 59 (20.9%) |
| Reduce workload by increasing manpower | 18 (6.4%) |
| Adequate rest/leaves/holidays | 14 (5.0%) |
| Proper duty roster/pre-decided work hours | 8 (2.8%) |
| Special COVID-19 allowance/monetary/transport and other benefits | 8 (2.8%) |

whereas 13% of responders had concerns about the infrastructure development and capacity building, and 10.1% worried about the availability of medical equipment and ICU beds. Availability of cheap, easy, and fast diagnostic kits (7.7%), more testing centres at private hospitals or every district hospital(6.3%), and staff training (4.3%) were also among the recommendations. Nearly 1.4% of the respondents suggested providing transportation facilities for HCWs, pre-operative screening of patients and providing dedicated ambulances to improve patient's access to hospitals. In the administration-related needs (n = 82), availability of the workforce (20.3%), rate enhancement of COVID-19 and other healthcare packages under Pradhan Mantri Jan Arogya Yojana(PMJAY) (6.3%); the National Health Insurance Scheme; monetary and administrative support to hospitals from the government (4.8%), availability of teleconsultation (1.9%), and better public-private coordination (1.9%) were among the major domains according to hospitals where urgent attention was required. Of other administration-related suggestions, 1% each suggested allotting dedicated staff in hospitals to coordinate with the government health insurance authorities, provide a transparent evaluation of cases under PMJAY and insist on the need for strategization to prevent further spread. 0.5% each suggested developing an online record system for all hospitals, providing e-tokens through mobile apps and extending PMJAY cover to more citizens based on partial payment. In the HCW care-related needs (n = 30), safety and overall well-being were the top concerns (8.7%), which included fixed working hours, timely salary disbursement, and better handling of conflicts with patient attendants. It was followed by the provision of insurance, allowances, and other benefits to HCWs (4.8%) and social security to HCWs and their families (1%). In the patients' care- and pubic-related needs (n = 29), public awareness (6.3%), simplified guidelines from the government (4.3%) and counselling of COVID-19 suspected patients at non-COVID-19 hospitals (1.9%) were among the top concerns of the responders. Among other suggestions (n = 2), 0.5% each emphasized that even hospitals designated as COVID-19 hospitals had no COVID-19 labs and district hospitals were refusing their samples which was impacting non-COVID-19 patients' care.

**3.5.1. Model 1—data analysis.** We used the variables listed in Fig 1 to train our regression model, and calculated the RMSE scores for the seven output variables as summarized in Table 4. By analyzing the RMSE scores, healthcare providers and policymakers can understand the factors that affect the different healthcare services and surgeries during the COVID-19 pandemic.

The "Obstetric Services Patients" had the lowest RMSE of 0.20, which indicates that it has a better association with the input variables.

Larger hospitals (>250 beds) showed a stronger correlation with these output variables compared to smaller hospitals, indicating that hospital size is an important factor to consider when examining healthcare facilities and their capabilities summarized in Fig 1.

**Table 4. Root Mean Squared Error(RMSE) for the seven output variables in Model 1 for a joint regression model trained on all the input variables.**

| Output Variable | RMSE Score |
| --- | --- |
| Emergency Patients | 0.31440 |
| Obstetric Services Patients | 0.19446 |
| OPD Patients | 0.24817 |
| Emergency Surgeries | 0.35730 |
| Obstetric Services Surgery | 0.23600 |
| OPD Surgeries | 0.24016 |
| Elective Surgeries | 0.25302 |

The correlation coefficient between "hasScreeningForCOVID-19" and "hasSeperateFacility-ForCOVID-19" is 0.43, indicating a moderate positive correlation between these two variables. This means that healthcare facilities that have screening for COVID-19 are also more likely to have a separate facility for COVID-19 patients.

Regarding infrastructure, larger hospitals with over 100 beds were weakly positively correlated with having separate screening and triaging areas for COVID-19 patients as well as separate facilities for COVID-19 patients and were more likely to have negative pressure rooms. The correlation slightly increased for hospitals with over 250 beds. On the other hand, small hospitals with less than 50 beds were the least likely to have separate facilities for COVID-19 and non-COVID-19 patients, with a correlation coefficient of -0.5. Large hospital systems were weakly positively correlated with having an adequate supply of personal protective equipment (PPE).

In terms of type of facility, large public hospitals (100–250 bed capacity) were more likely to have screening and triaging facilities for COVID-19 patients. They had policies for staff safety and testing in place. They were most unlikely to face management challenges like unavailability of staff during COVID-19, sufficient essential supplies (PPEs) for emergency surgeries and adequate staff training.

Regarding surgery, emergency surgery was more likely in hospitals which had enough supply of protective equipment like PPE and N95 masks.

**3.5.2. Model 2 to measure staff and revenue decline.** We used the relevant variables listed in Fig 2 to calculate the decrease in staff and revenue. The calculated RMSE scores for the five output variables are summarized in Table 5.

Therefore, based on the RMSE scores, the input variables have a stronger relationship with reduction in revenue, especially significant with % reduction in revenue"40–49% and > 50%" output variable, and the model is better able to predict this output variable compared to the other four output variables. This information can be valuable for healthcare professionals and policymakers as it helps them comprehend the factors that lead to a decline in staff and revenue during the COVID-19 outbreak. With this understanding, they can develop effective strategies to minimize the negative impact of these factors.

Model 2 depicts that public hospitals and large (>250 bed) private not-for-profit hospitals were seen to have a very low correlation with staff shortages and a decrease in revenue, whereas private for-profit hospitals(<100 beds) had a reduction in revenue and staff (Fig 2).

To investigate the association between our input variables and output variables, we conducted a Fisher's exact test after generating a correlation matrix. By performing this test, we were able to obtain p-values that helped us identify significant relationships. We represented the relationship between the input and output variables along with their respective p-values in the graphs below.

To further enumerate significant variables we used bar plots that revealed emergency surgery was provided by hospitals with separate COVID-19 facilities, N95 masks, and a sufficient

**Table 5. Root mean squared error for the five output variables in Model 2 for a joint regression model trained on all the input variables.**

| Variables | RMSE Score |
|---|---|
| Reduction in staff | 0.4618 |
| % reduction in revenue | 0.3925 |
| <10% and 10–19% (% of reduction) | 0.5392 |
| 20–29% and 30–39% (% of reduction) | 0.4864 |
| 40–49% and >50% (% of reduction) | 0.3207 |

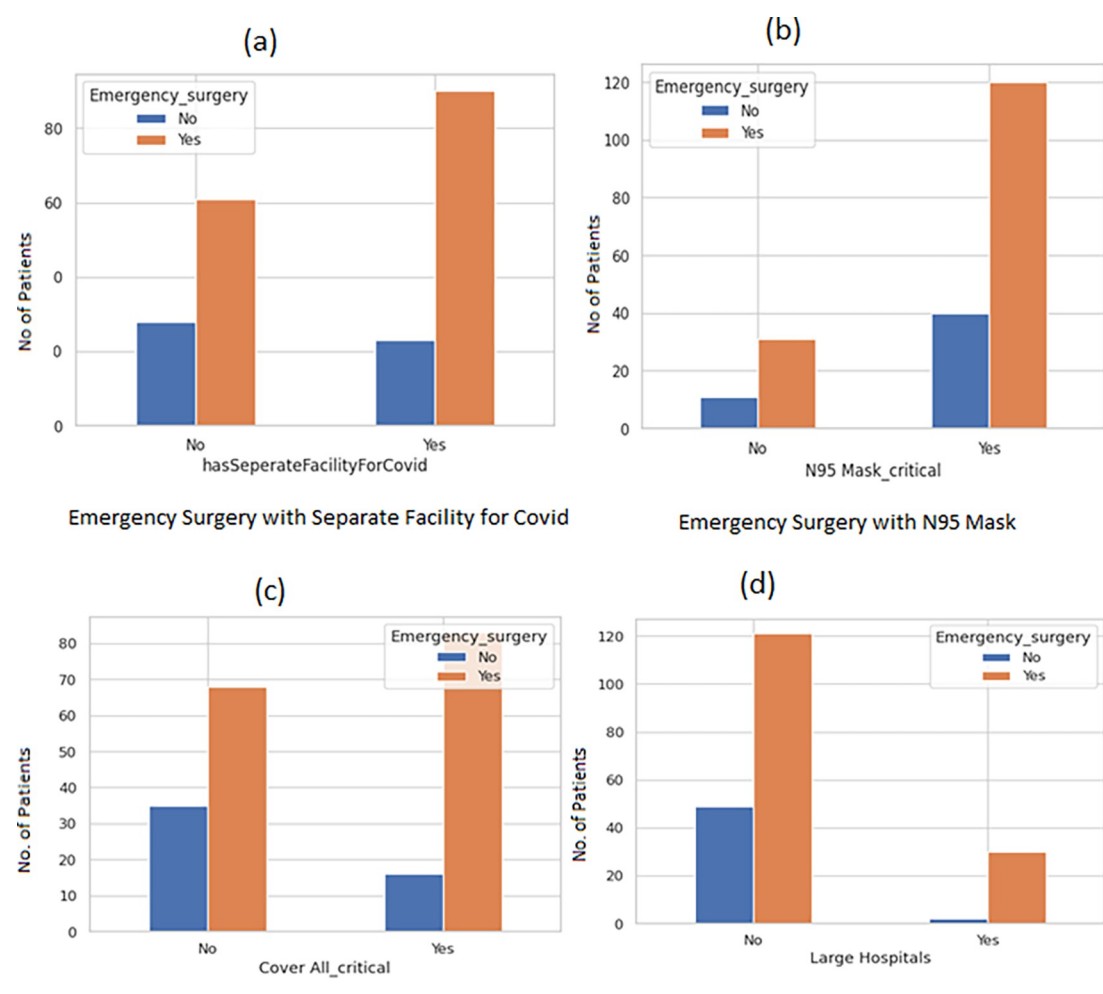

**Fig 3. Bar plot to depict the relationship of emergency surgeries with four input variables.**

supply of PPE (Fig 3A–3C). Emergency surgery was more likely to be continued in larger hospitals >250 beds (Fig 3D)

Large hospitals with >250 beds had separate facilities for COVID-19 patients. The bar plot (Fig 4A) is suggestive of the significance with a value-value < 0.001. Small hospitals were unlikely to have a separate facility for COVID-19 p value < 0.001 (Fig 4B). Elective surgeries were more likely to take place in hospitals with less than 50-bed capacity. The p-value for the same was 0.00050 (Fig 4C).

Public hospitals and large (>250 bed) private for-profit hospitals were least likely to face staff shortages (Fig 5A) and a decrease in revenue (Fig 5B). Whereas private not-for-profits had a significant reduction in staff (Fig 5C) with a p-value of 0.0028. There was a significant correlation reduction in staff p-value 0.0316) with seven INSIGHT variables (stress and anxiety of HCWs) (Fig 6). It was also seen that adequate essential supplies (PPEs) in critical areas were present in large private and Public (hospitals (Fig 7A and 7B). There was a significant correlation of Ayushman Bharat–Pradhan Mantri Jan Arogya Yojana with Emergency surgeries (Fig 8) p-value < 0.001.

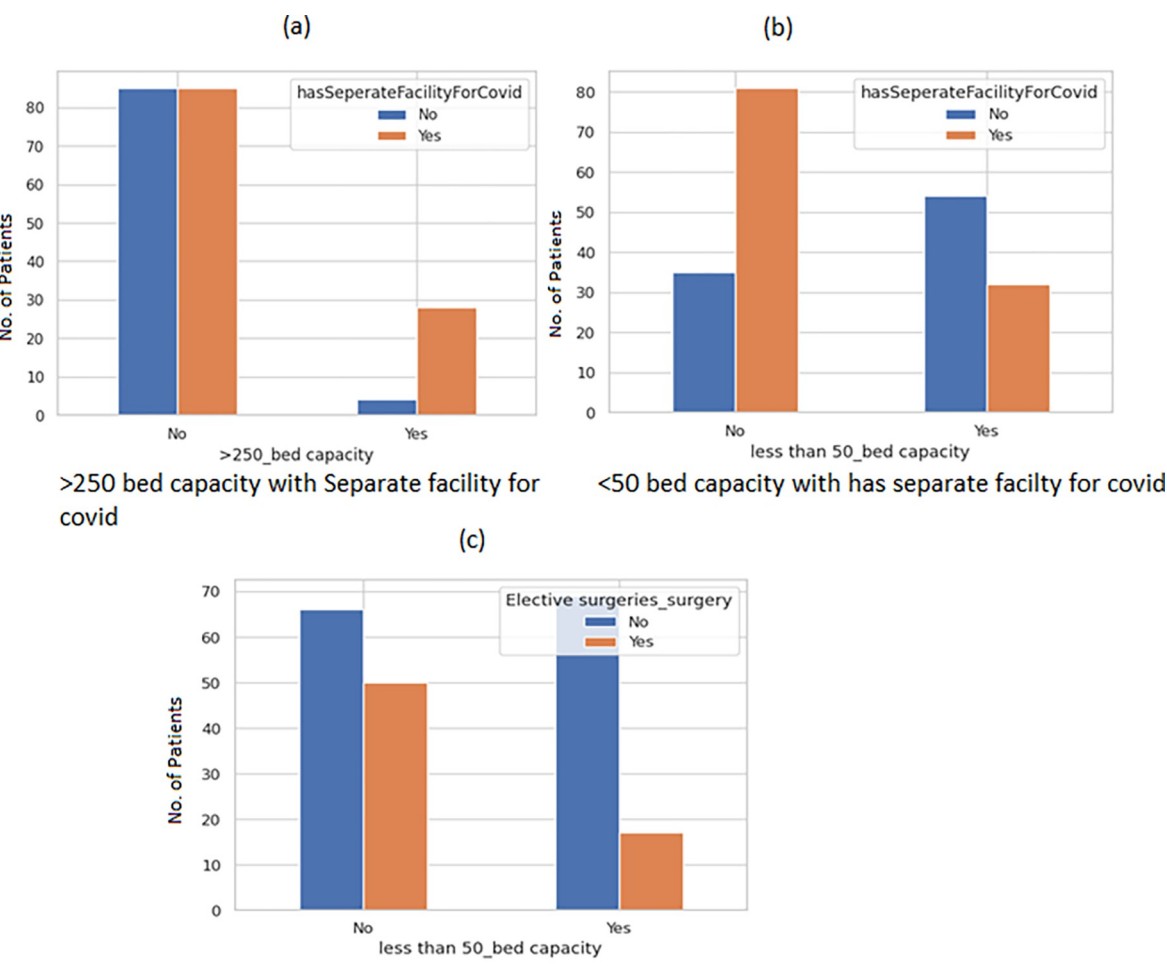

**Fig 4. Bar plot to depict the relationship of different bed capacities with variable has separate facility for COVID.**

## 4. Discussion

This survey was carried out between May and July 2020, when the pandemic was fast spreading.

The present study offers an overview of the various obstacles encountered in hospital settings, as well as the strategies and measures implemented to effectively address this unparalleled predicament. The survey encompassed a cohort of individuals who held positions as administrators or senior health professionals, assuming administrative duties within their respective roles. The primary objective of our study was to analyze the various factors that contributed to the successful maintenance of crucial hospital operations and the provision of care to patients affected by the pandemic. Additionally, we aimed to identify the specific elements that contributed to the heightened stress levels experienced by healthcare professionals during this challenging period.

### 4.1. Decline in hospital services and surgeries

This nationwide survey demonstrates a significant decline in essential healthcare services in both outpatient and inpatient departments. Our findings are consistent with the studies

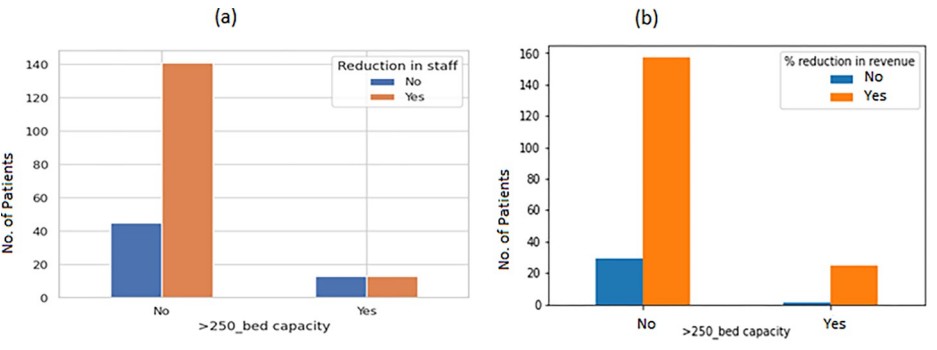

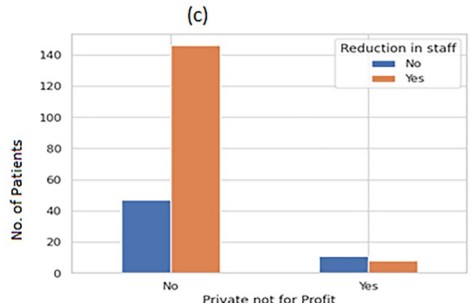

**Fig 5. Bar plot to show the relationship of reduction in staff percentage reduction in revenue with bed capacity and the type of facility.**

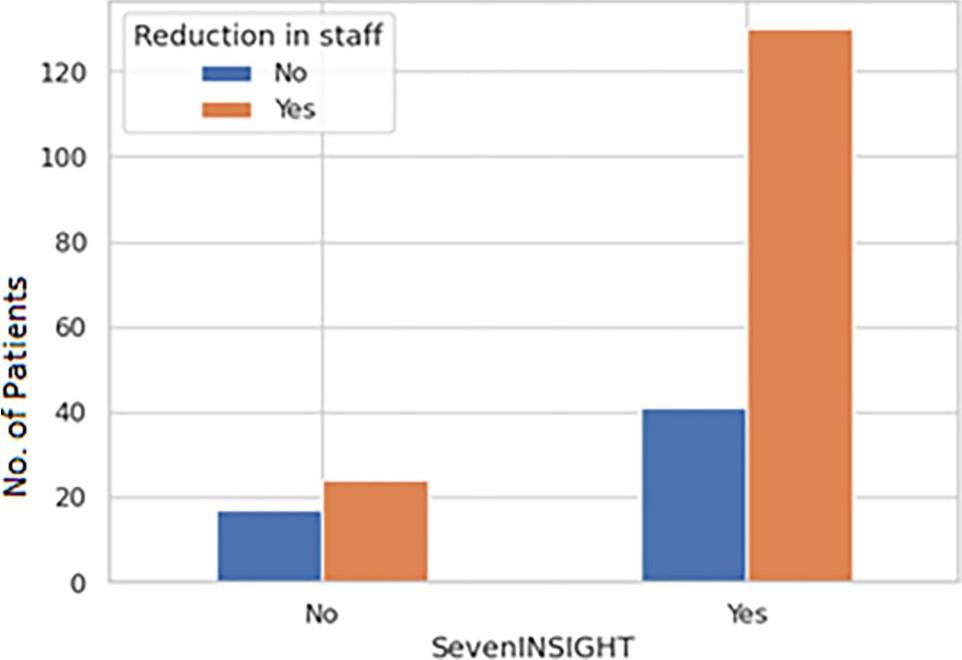

**Fig 6. Reduction in staff with sevenINSIGHT.**

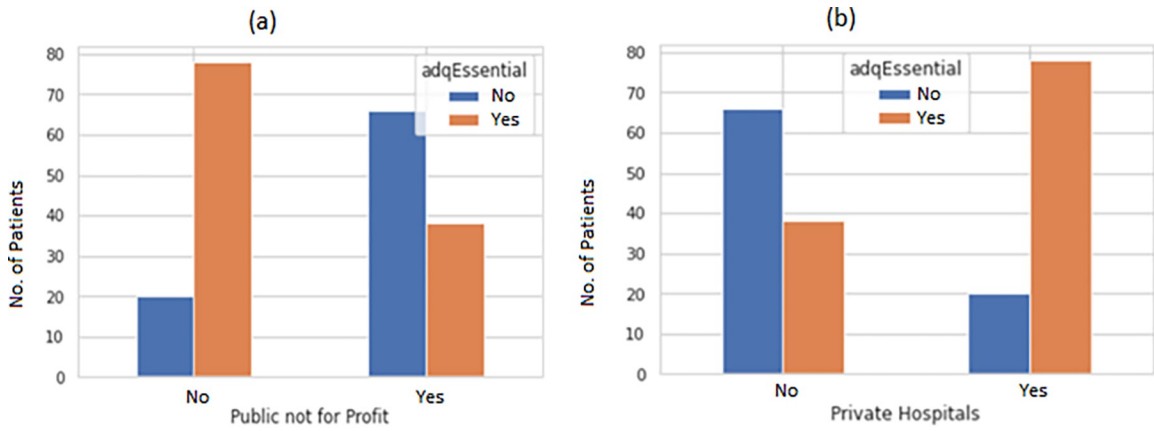

**Fig 7. Bar plot to show the relationship of adequate essential with type of facility.**

conducted by Yamaguchi et al. in November 2020 [6] and Zemzem Shuka et al. in February 2022 [7], and slightly lower than the research conducted by P Chatterji et al. in January 2021 [8] and Engy et al. in July 2020 [9]. According to this nationwide study, 56.2% of hospitals did not offer obstetric services, and 63.3% did not provide obstetric procedures throughout the pandemic. These findings align with a study conducted by J. Jardine et al. in April 2021 [10], however, the percentages are slightly lower compared to a study by Fahmy et al. in November 2021 [11]. Our study illustrates a noteworthy decrease in the utilisation of emergency services by 33.6% in hospitals, as well as a reduction of 44.5% and 77.4% in emergency and elective surgeries, respectively. These findings are consistent with the research conducted by Stanley Xu

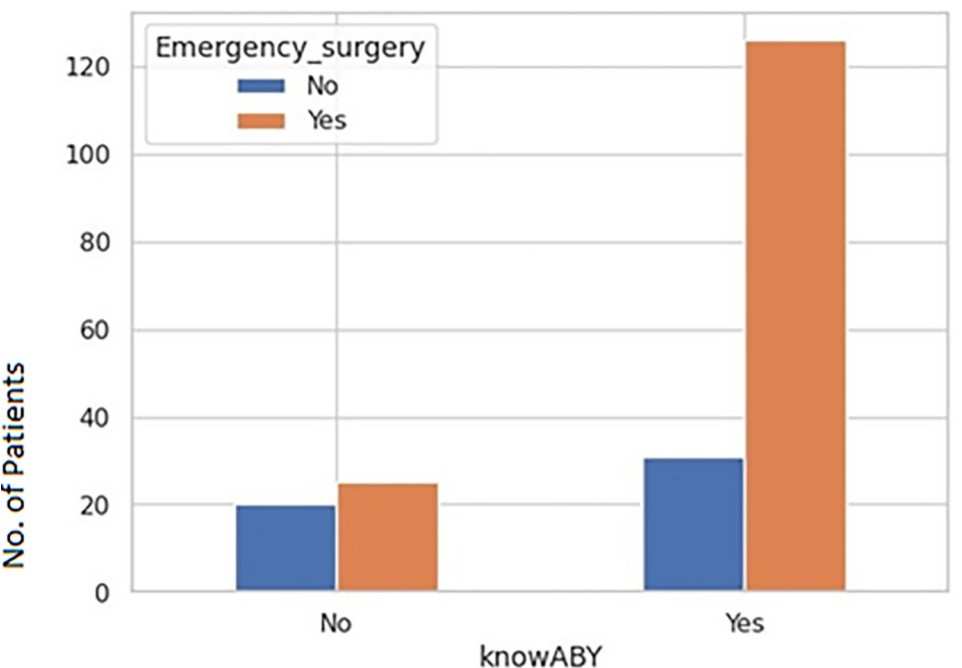

**Fig 8. Bar plot to show the relationship of knowABY with type of emergency surgeries.**

et al [12] in April 2021, Martin Hübner et al in October 2020 [13], Ahmet Surek et al in November 2020 [14], and Thomas D Dobbs et al in June 2021 [15].

## 4.2. Infrastructure and type of facility

It was seen that large hospitals with> 250 beds were more likely to have COVID-19 triaging and screening areas, separate facilities, and negative pressure for COVID-19 patients. The study by Gupta et al found that patients who were admitted to hospitals with fewer than 50 ICU beds had a more than threefold higher risk of death than patients admitted to larger hospitals [3]. This was harder to replicate in smaller hospitals. Additionally, hospitals that successfully segregated COVID-19 patients in dedicated areas were able to continue offering non-COVID-19 care. Consequently, larger private hospitals demonstrated a higher tendency to deliver emergency care, including emergency surgeries and elective procedures [14]. By isolating those affected, HCWs could better organize and deliver necessary care to both COVID-19 and non-COVID-19 patients. On the other hand, larger public hospitals became overwhelmed with COVID-19 care and potentially could have enlisted private contract staff to cater to non-COVID-19 cases. In response to the situation, extensive hospital networks also initiated tele-consultation services. However, many smaller private and physician-operated hospitals were forced to close due to elective procedures being put on hold and the inability to have a separate area to segregate COVID-19 patients. Hence to ensure, the highest possible utilization of critical resources to best meet this expected pandemic surge and non-COVID-19 care, the adoption of a digital state-wide tracking system for beds, ventilators and facilities is imperative. Resource allocation in silos would be detrimental and delay access to the right level of care.

In future, if such situations arise, it is of paramount importance that the government and society come together as one to support HCWs who are literally and metaphorically, soldiers on invisible battle lines. Counselling and therapy helplines should be available round the clock for HCWs in such unprecedented times.

The healthcare infrastructure must continue functioning smoothly. There should be a combined, coordinated effort from the government and private sector to proceed with services with the least road bumps possible. For this, guidelines must be formulated well in advance with clear and specified roles during such times.

By increasing the number of healthcare facilities at the grassroots, or expanding the small and medium-sized facilities, medical care can be made more accessible to people living in remote areas. It will also reduce the massive burden of cases faced by tertiary hospitals during these dire circumstances, thus providing relief to both the patients and the HCWs.

It will be more pragmatic to equip such facilities for pandemic-like eventualities, beforehand. Since these hospitals are of a smaller scale and are better integrated into the communities they serve, there will be increased trust between the HCWS and patients, aiding in the seamless spread of messages of public health importance.

Preparing early for such times will aid in developing programs for emergencies which can be implemented quickly, with a lesser overhead cost for both the facility as well as the government. Thus, the diverse needs of the population at the grassroots level can be met.

## 4.3. Staff shortage and safety of HCWs

The nationwide survey reveals a staff shortage of 50.5% across multiple roles (administrative staff, nurses, doctors), alongside a significant 49.5% reduction in Intensive Care Unit personnel which is lower than the findings from Huiwen Xu et al., Oct 2020 [16], Batra et al., Dec 2020 [17] and Jackie Nguyen et al., Oct 2021 [18]. The role of HCWs as frontline responders during a pandemic is of paramount importance. Fear of contracting the virus, spreading it to

their families, and ensuring that their children were cared for with schools being shut, weighed heavily on their minds [19]. Hospitals that implemented policies for staff testing, covered the cost of testing, and conducted risk assessments observed the least decline in staff numbers. Survey participants recommended postponing non-essential surgeries, increasing COVID-19 testing, regularly screening patients, periodically testing HCWs for COVID-19, ensuring sufficient PPE supplies, and establishing suitable quarantine facilities. For comprehensive well-being, beyond attending to the physical and mental needs of HCWs, financial stability should also be ensured. This can be achieved by offering medical and life insurance, special COVID-19 allowances (e.g., transportation), and providing essential supplies for HCWs and their families during these challenging times [20]. Furthermore, proposed policy recommendations aim to ensure a fair distribution of healthcare workers (HCWs) between COVID-19 and non-COVID-19 patients, coupled with an expansion in HCW recruitment. Considering the loss of lives among HCWs globally, respondents expressed palpable stress. To address this, ongoing motivation from the government and the public is crucial [21]. Establishing neuropsychiatric helplines, counselling sessions, and promoting practices like yoga and pranayama were proposed to alleviate stress. A meta-analysis demonstrated that yoga is particularly effective in reducing occupational stress among various relaxation techniques [22]. Additionally, motivational support from authorities and the public enhanced the sense of protection and value, significantly improving morale as they faced stigmatization at the workplace and in society [21].

## 4.4. Reduced essential supplies

It is imperative during a pandemic that a global supply chain be maintained. Reduced essential supplies during this time can result in drastic consequences, with a severe impact on public health.

The unavailability of hospital equipment such as ventilators, PPEs, and oxygen cylinders can lead to a steep increase in mortality rates and other complications. A limited number of kits for screening can hamper attempts to curb disease spread.

The scarcity of vaccines and drugs in the supply chain results in excessive price rises, furthering the socio-economic divide and causing financial strain even for the affording.

A study called attention to resource shortages in low-income countries like Ethiopia which inhibited effective medical treatment of COVID-19 afflicted patients. [23]

Adverse effects of disrupted supply chains were felt even in high-income countries such as the USA, as the pandemic exposed gaping holes in the healthcare preparedness of developed countries, too. [24]

Numerous other studies [25–28] highlighted the disastrous consequences of reduced supplies during the pandemic and the urgent need to overcome the stop-gap arrangement for global pandemics.

## 4.5. Health insurance and policies

Our study indicates a positive correlation between hospitals performing emergency surgeries in patients covered under the PMJAY scheme. A study from Figueroa et al. [20] also revealed that low-income patients with government (Medicaid) insurance had better access to health care.

During a pandemic, patients must have access to necessary medical services, including screening and confirmatory tests, medication, and surgeries. This leads to significant expenditure, increasing the amount of medical debt burden faced by families. In such times, insurance provides a safety blanket, reducing the catastrophic financial distress faced by the populace. Early and effective treatment of the disease in the common people will lead to curbing the

spread and better containment of the disease, and the residual monetary resources can be used on other essential needs.

Expansion of health insurance coverage coverage can address the income disparities affecting the grassroots' citizens. It bridges the gap, thus ensuring that the impoverished and marginalized people receive the necessary care.

It also bolsters the economy of the country, leading to a quicker recovery of the workforce and the market, as a healthier population is more productive.

Therefore, we recommend insuring maximum people before and during a pandemic.

### 4.6. Conclusions and lessons learned

Numerous hospitals ceased providing surgical services during the pandemic, emphasizing the importance of safeguarding surgical ecosystems to ensure comprehensive care for the general population. It is imperative that emergency and obstetric services continue [15], and obstetric services must continue across a majority of hospitals, prioritizing the safety of surgeons and HCWs [29]. Leveraging these healthcare systems could effectively serve cancer patients, as well as those requiring medical and trauma surgeries, as delays could lead to disease progression and unfavourable survival outcomes. Small speciality hospitals (SSH) could address this situation by implementing gate-side screening and directing COVID-19 patients to designated hospitals, if feasible. Efficiently triaging patients in COVID-19-designated hospitals and establishing COVID-19-free facilities for non-COVID-19 surgical, obstetric, and elective services necessitates a unified state-wide data-driven system that streamlines patient transfers. Having insights into bed occupancy, patient load, and facility availability would expedite transfers and treatments while optimizing existing resources to the best possible pandemic scenarios, governmental collaboration with all stakeholders, including private hospital administrators and consultant bodies, is vital to ensure a cohesive approach to pandemic management without completely halting non-COVID-19 services. Expanding the coverage of AB PM-JAY to more hospitals and the population would bolster especially small speciality hospitals (SSH) capabilities. For non-urgent surgeries, hospitals should consider patients' medical needs, logistical capacities, and real-time risks. The medical necessity of a procedure should be evaluated by a surgeon with expertise in the relevant field to assess potential medical risks due to cause delays. Administrative staff should determine logistical feasibility, considering hospital and community limitations and factoring in facility resources (beds, staff, equipment, supplies, etc.) as well as community safety and well-being.

## Supporting information

**S1 Data.**
(XLS)

## Author Contributions

**Data curation:** Mona Duggal, Kangan Maria.

**Formal analysis:** Mithlesh Chourase, Mukesh Kumar.

**Funding acquisition:** Bhanu Duggal.

**Investigation:** Bhanu Duggal.

**Methodology:** Kangan Maria, Vasuki Rayapati.

**Project administration:** Bhanu Duggal.

**Resources:** Bhanu Duggal.

**Supervision:** Mona Duggal, Praveen Gedam, Lakshminarayanan Subramanian.

**Visualization:** Sujata Saunik.

**Writing – original draft:** Mona Duggal, Mukesh Kumar.

**Writing – review & editing:** Anuva Kapoor, Vasuki Rayapati.

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
