## [Decision Letter · Decision Letter 0]

3 Jan 2023

PGPH-D-22-01896

Health Care Delivery and Keeping the Workplace Safe in the COVID Pandemic: Results of a Cross-Sectional Survey of Hospitals Delivering Care Under the National Health Insurance Scheme in India

Dear Dr. Kumar R,

Thank you for submitting your manuscript to PLOS Global Public Health. After careful consideration, we feel that it has merit but does not fully meet PLOS Global Public Health’s publication criteria as it currently stands. Therefore, we invite you to submit a revised version of the manuscript that addresses the points raised during the review process.

We look forward to receiving your revised manuscript.

Kind regards,

Giridhara R Babu, MBBS, MPH, PhD

Academic Editor

Journal Requirements:

2. Please provide separate figure files in .tif or .eps format only and remove any figures embedded in your manuscript file. Please also ensure that all files are under our size limit of 10MB.

3. Tables should not be uploaded as individual files. Please remove these files and include the Tables in your manuscript file as editable, cell-based objects. For more information about how to format tables, see our guidelines:

https://journals.plos.org/globalpublichealth/s/tables

4. We have noticed that you have uploaded Supporting Information files, but you have not included a list of legends. Please add a full list of legends for your Supporting Information files after the references list. 

5. In the online submission form, you indicated that "Data Availability Statement: The authors confirm that the combined aggregated data from our study can be made available upon request. For assistance in obtaining access to the data, please contact bhanuduggal2@gmail.com". All PLOS journals now require all data underlying the findings described in their manuscript to be freely available to other researchers, either 1. In a public repository, 2. Within the manuscript itself, or 3. Uploaded as supplementary information.

Additional Editor Comments (if provided):

Reviewers' comments:

Reviewer's Responses to Questions

**Comments to the Author**

1. Does this manuscript meet PLOS Global Public Health’s publication criteria? Is the manuscript technically sound, and do the data support the conclusions? The manuscript must describe methodologically and ethically rigorous research with conclusions that are appropriately drawn based on the data presented.

Reviewer #1: Yes

Reviewer #2: Yes

Reviewer #3: Yes

Reviewer #4: Yes

Reviewer #5: Yes

2. Has the statistical analysis been performed appropriately and rigorously?

Reviewer #1: Yes

Reviewer #2: I don't know

Reviewer #3: Yes

Reviewer #4: I don't know

Reviewer #5: Yes

3. Have the authors made all data underlying the findings in their manuscript fully available (please refer to the Data Availability Statement at the start of the manuscript PDF file)?

Reviewer #1: Yes

Reviewer #2: Yes

Reviewer #3: Yes

Reviewer #4: Yes

Reviewer #5: Yes

4. Is the manuscript presented in an intelligible fashion and written in standard English?

Reviewer #1: Yes

Reviewer #2: Yes

Reviewer #3: Yes

Reviewer #4: No

Reviewer #5: Yes

5. Review Comments to the Author

Reviewer #1: Health Care Delivery and Keeping the Workplace Safe in the COVID Pandemic: Results of a Cross-Sectional Survey of Hospitals Delivering Care Under the National Health Insurance Scheme in India

My sincere appreciation to the Editor, PLOS Global Public Health for consideration of being a reviewer of the above-named manuscript. This is well appreciated.

The manuscript highlights delivery of healthcare services and safety of healthcare workers during the COVID-19 pandemic period in India across health facilities under the National Health Insurance Scheme. The National Health Insurance Scheme globally is believed to be part of governments efforts in ensuring healthcare services are available, accessible, and affordable, thereby reducing out of pocket expenditure on healthcare services.

The authors research work on this topic is timely and of importance considering the global impact of COVID-19 pandemic on provision of healthcare services across all tiers of healthcare. Authors are to address the following comments as follows:

General comments

1. There are abbreviations used in the manuscript text that were not firstly defined before being used. Authors should address this observation.

2. Authors need to be consistent with the use of COVID-19, and not COVID

Title

1. Need for the full title to be revised based on the number of words

2. COVID in the title should be COVID-19

Abstract

Findings to be changed to ‘’Results’’

Introduction

1. Need for the introduction section to be reframed. The format used by the authors does not follow the normal flow of background information, magnitude of the problem, rationale or justification for the survey, and survey objectives.

Materials and Methods

1. This section needs to be well structured, with the sequence of study area, study design and duration, study population and eligibility criteria, sample size determination, sampling technique, data collection method(s), data analysis

2. There was no proper description of the survey area in terms of states and health facilities highlighted under the results section

3. How were the health facilities selected? Authors should state criteria used for selection of Public and Private health facilities

4. Authors to clearly state who the study population were, eligibility criteria used for selection?

5. The survey sample size and how it was calculated? Was proportional allocation done for selection of respondents from public and private health facilities?

6. Authors should highlight the sampling technique used for the survey.

7. There was no information on public & private health facilities under the National Health Insurance Scheme in India

8. There was no description of health care services provided under the National Health Insurance Scheme in India

9. Information regarding urban hospitals to be completed before that of rural hospitals having poor internet

10. ‘’In case of rural areas where the hospitals often experience poor internet connectivity, the research team helped them in filling up the form through telephone interviews’’. How did the researchers address response bias? For responses that did not meet the interviewers’ expectations, how was this handled? Were there some probing questions used during the interview?

11. The type of data analysis done by authors not clearly illustrated. Level of statistical significance not documented.

12. ‘’We used logistic regression is used to understand the effect of hospital beds……’’. Delete ‘’is used’’

13. Operational definitions should be included by authors

14. Authors did not define what dependent (outcome) variables used, same for the independent (explanatory) variables

15. ‘’In this dataset, we considered 18?? variables’’. Authors are to clarify this.

Results

1. It will be good that authors should include absolute numbers alongside proportions being reported under the results section

2. Authors need to include the response rate as part of the first sentence in the first paragraph. (Section 3.1: State-wise distribution of the responses) Authors are to provide information on the sample size under the methods from which they will be able to provide information on the response rate

3. ‘’At least 5% of all empaneled hospitals under Ayushman Bharat Yojana were expected to respond from each state’’. This sentence under Lines 2-3, Section 3.1 supposed to be under Methods section.

4. Authors should provide information on what constitute ‘’others (8.9%)’’ line 3 under the Section 3.2 Demographic characteristics of responding hospitals

5. Line 5 under Section 3.3 Changes in hospital services and availability of healthcare workers (‘’55.1% hospitals reported a decrease in number of OPD patients visiting…….). Authors used numbers to open a sentence, this should be corrected. Same for line 8 (‘’10.5% hospitals responded to the scarcity of nurses…) and line 13 (‘’49.5% were experiencing scarcity of staff in critical non-COVID areas such as intensive care units and dialysis units)

6. Section 3.3 Changes in hospital services and availability of healthcare workers Lines 11-13 (‘’It shows the extent of the scarcity of HCW in both private and public hospitals across all covered states’’) should be moved under the Discussion section.

7. Authors need to provide information on COVID and non-COVID areas under the methods section

8. Section 3.4 Determinants of hospital services. Authors provided information on bivariate analysis findings; however, this was not captured under the methods section.

Discussion

The authors at the opening of the discussion section should first provide overall summary findings of the survey by providing information that broadly answers the survey objectives.

Tables

1. Incomplete title for all the tables. Tables are meant to be self-explanatory. Authors supposed to reflect what, when, and where?

2. All tables need to be well formatted for the frequencies and percentages to align.

Figures

1. Incomplete title for all the figures. Authors supposed to reflect what, when, and where?

2. Figure 2, axes are not labelled

3. Model (Fig 3), no title or axis titles

4. Model 2 (Fig 4), no title or axis titles

Reviewer #2: The authors have done a commendable job analyzing data to evaluate healthcare delivery, safety, and various problems experienced during the ongoing pandemic. However, it will be amazing if the authors can address the following minor concerns:

a) A description of the Ayushman Bharat Pradhan Mantri Jan Arogya Yojana and an overview of national insurance schemes operating shall be added in the introduction.

b) A comparison of the current deficiencies (ICU, staff, equipment, etc) with the pre-pandemic times can help readers develop a better understanding.

c) Any limitations within the study model must be included too.

Reviewer #3: This study is of significant to India as it gives insight ino the availability of manpower and infrastructure in health sector in India during pandemic and areas that need improvement in case of future pandemic

Reviewer #4: Thank you for providing me the opportunity to review this manuscript. The authors have provided evidence on a significant topic, i.e. of healthcare delivery system in India during the COVID-19 pandemic. My comments are attached in the file.

Reviewer #5: The paper was intelligently written. It was quite easy to follow and understand given the choice of words used.

There are a few grammatical errors that may need to be corrected in most of the sessions.

The response to the question on the competing interests is not quite clear. It is important also to highlight clearly the problem in the background section of the abstract

6. PLOS authors have the option to publish the peer review history of their article (what does this mean?). If published, this will include your full peer review and any attached files.

**Do you want your identity to be public for this peer review?** For information about this choice, including consent withdrawal, please see our Privacy Policy.

Reviewer #1: No

Reviewer #2: No

Reviewer #3: No

Reviewer #4: No

Reviewer #5: No

---

## [Decision Letter · Decision Letter 1]

17 Jul 2023

PGPH-D-22-01896R1

Cautionary lessons from the Covid-19 Pandemic : Healthcare systems grappled with the dual responsibility of delivering Covid and non-Covid care.

Dear Dr. Duggal,

Thank you for submitting your manuscript to PLOS Global Public Health. After careful consideration, we feel that it has merit but does not fully meet PLOS Global Public Health’s publication criteria as it currently stands. Therefore, we invite you to submit a revised version of the manuscript that addresses the points raised during the review process.

We look forward to receiving your revised manuscript.

Kind regards,

Giridhara R Babu, MBBS, MPH, PhD

Academic Editor

Journal Requirements:

Additional Editor Comments (if provided):

Reviewers' comments:

Reviewer's Responses to Questions

**Comments to the Author**

1. If the authors have adequately addressed your comments raised in a previous round of review and you feel that this manuscript is now acceptable for publication, you may indicate that here to bypass the “Comments to the Author” section, enter your conflict of interest statement in the “Confidential to Editor” section, and submit your "Accept" recommendation.

Reviewer #4: (No Response)

Reviewer #5: (No Response)

2. Does this manuscript meet PLOS Global Public Health’s publication criteria? Is the manuscript technically sound, and do the data support the conclusions? The manuscript must describe methodologically and ethically rigorous research with conclusions that are appropriately drawn based on the data presented.

Reviewer #4: Partly

Reviewer #5: Yes

3. Has the statistical analysis been performed appropriately and rigorously?

Reviewer #4: I don't know

Reviewer #5: Yes

4. Have the authors made all data underlying the findings in their manuscript fully available (please refer to the Data Availability Statement at the start of the manuscript PDF file)?

Reviewer #4: Yes

Reviewer #5: Yes

5. Is the manuscript presented in an intelligible fashion and written in standard English?

Reviewer #4: No

Reviewer #5: Yes

6. Review Comments to the Author

Reviewer #4: It seems like the authors have addressed the concerns raised by reviewer 1 only (as seen in the response to reviewers document). While it may cover most of the concerns, the authors are requested to check the comments provided by other reviewers as well to improve their paper since it is currently lacking the criteria to be published. The typographical errors are still there which should be corrected as well.

Reviewer #5: In this era where the world experiences a rapid change in disease epidemiology and landscape, this topic is vert relevant. After a thorough review, I would like to bring to the attention of authors, the following things that need to be addressed:

Abstract: The background should be expanded to include more details especially the problem that the study intends to address.

Background: Some citations are missing, and authors must ensure the section provides a strong basis for the variables that are discussed in the methods, results and discussion eg., established impacts of the pandemic on hospital revenues, staffing levels etc. This provides a clarity for the choice of possible confounders.

Materials and methods

The section is detailed however, it raises questions on why some variables were included because no prior explanation was given in the background. Also, explanation is needed on how the questionnaire was able to explicitly capture the following; risk assessment, hospital staff knowledge about insurance schemes, and what was the reference period to compare changes in staffing levels

Results

General: for details before tables should be summarized, concise and only most striking features should be narrated. The rest will be seen on the tables.

Consistency: the % (percentage) could be placed on the topmost row of respective column of each table and not across each result on the table. In results section, interpretation/discussion is not expected unless when results and discussion section are merged together, which is not the case here.

It is important to have more subsections under the results sections for instance, separating results for the two models.

The statements, one: Large private for profit hospitals were less likely to face staff shortages and second; private for profit hospitals had significant staff reduction, are contradicting each other.

Figure 4 is not very clear i.e., what do the horizontal bar labels represent? For instance for the graph focused on beds > 250 does one represent those less than 250 and 1 those above it? Also, for private not for profit, what does 1 and 0 represent. All graphs miss labels on the vertical axis, please include them.

It is also important for authors to provide, in the methods section, the hypothesis behind the the choices of variables or associations For example, does high/low bed capacity represent or is it used as a proxy for hospital financial capital? Otherwise, how do we provide a linkage between bed capacity and presence of separate Covid facility or increase and decrease in non-covid hospitalizations?

Comparatively and in terms of proportion, bars with labels 0 for both large and small hospitals, show that majority of these had separate facility for Covid and offered more elective surgeries than those labelled 1. This was not explained though.....!

Discussion

This section requires a thorough revision

1. Reduce/avoid repeating the results. Instead authors should focus to interpret/show implication and discuss the results.

2. Discussion partly entails showing and interpreting similarities and differences of the current study and others which were done elsewhere. This was hugely missing. Very few comparisons were made.

3. Some intext citations were missing and should be included. At some points, citations were put at the end of sections which wholly contained the results of the current study. This is not proper and is unacceptable.

4. Ethical approval and IRB statements are same and one should be dropped.

References

Revision of some references is needed as they don't follow the format eg 16-18

7. PLOS authors have the option to publish the peer review history of their article (what does this mean?). If published, this will include your full peer review and any attached files.

**Do you want your identity to be public for this peer review?** For information about this choice, including consent withdrawal, please see our Privacy Policy.

Reviewer #4: No

Reviewer #5: No

---

## [Decision Letter · Decision Letter 2]

31 Oct 2023

PGPH-D-22-01896R2

Healthcare  delivery and keeping the workspace safe in the COVID pandemic: Results of a cross sectional survey of hospitals in India for the shaping of policies for future pandemics.

Dear Dr. Duggal,

Thank you for submitting your manuscript to PLOS Global Public Health. After careful consideration, we feel that it has merit but does not fully meet PLOS Global Public Health’s publication criteria as it currently stands. Therefore, we invite you to submit a revised version of the manuscript that addresses the points raised during the review process.

We look forward to receiving your revised manuscript.

Kind regards,

Miquel Vall-llosera Camps

Staff Editor

Journal Requirements:

2. Please ensure that the Title in your manuscript file and the Title provided in your online submission form are the same.

Reviewers' comments:

Reviewer's Responses to Questions

**Comments to the Author**

1. If the authors have adequately addressed your comments raised in a previous round of review and you feel that this manuscript is now acceptable for publication, you may indicate that here to bypass the “Comments to the Author” section, enter your conflict of interest statement in the “Confidential to Editor” section, and submit your "Accept" recommendation.

Reviewer #5: All comments have been addressed

2. Does this manuscript meet PLOS Global Public Health’s publication criteria? Is the manuscript technically sound, and do the data support the conclusions? The manuscript must describe methodologically and ethically rigorous research with conclusions that are appropriately drawn based on the data presented.

Reviewer #5: Yes

3. Has the statistical analysis been performed appropriately and rigorously?

Reviewer #5: Yes

4. Have the authors made all data underlying the findings in their manuscript fully available (please refer to the Data Availability Statement at the start of the manuscript PDF file)?

Reviewer #5: Yes

5. Is the manuscript presented in an intelligible fashion and written in standard English?

Reviewer #5: Yes

6. Review Comments to the Author

Reviewer #5: Authors have worked on comments and this version shows significant improvement.

The revision of discussion subsection is not satisfactory.

1. There is repetition of results which is not advised.

2. Most results have not been discussed. Comparisons with findings from other studies were made but authors didn't provide an explanation behind the difference and the implications of survey results or their discrepancies with other findings was largely missing

7. PLOS authors have the option to publish the peer review history of their article (what does this mean?). If published, this will include your full peer review and any attached files.

**Do you want your identity to be public for this peer review?** For information about this choice, including consent withdrawal, please see our Privacy Policy.

Reviewer #5: No

---

## [Decision Letter · Decision Letter 3]

27 Feb 2024

PGPH-D-22-01896R3

Healthcare  delivery and keeping the workspace safe in the COVID pandemic: Results of a cross sectional survey of hospitals in India for the shaping of policies for future pandemics.

Dear Dr. Duggal,

Thank you for submitting your manuscript to PLOS Global Public Health. After careful consideration, we feel that it has merit but does not fully meet PLOS Global Public Health’s publication criteria as it currently stands. Therefore, we invite you to submit a revised version of the manuscript that addresses the points raised during the review process.

We look forward to receiving your revised manuscript.

Kind regards,

Miquel Vall-llosera Camps

Staff Editor

Journal Requirements:

Reviewers' comments:

Reviewer's Responses to Questions

**Comments to the Author**

1. If the authors have adequately addressed your comments raised in a previous round of review and you feel that this manuscript is now acceptable for publication, you may indicate that here to bypass the “Comments to the Author” section, enter your conflict of interest statement in the “Confidential to Editor” section, and submit your "Accept" recommendation.

Reviewer #5: All comments have been addressed

2. Does this manuscript meet PLOS Global Public Health’s publication criteria? Is the manuscript technically sound, and do the data support the conclusions? The manuscript must describe methodologically and ethically rigorous research with conclusions that are appropriately drawn based on the data presented.

Reviewer #5: Yes

3. Has the statistical analysis been performed appropriately and rigorously?

Reviewer #5: Yes

4. Have the authors made all data underlying the findings in their manuscript fully available (please refer to the Data Availability Statement at the start of the manuscript PDF file)?

Reviewer #5: Yes

5. Is the manuscript presented in an intelligible fashion and written in standard English?

Reviewer #5: Yes

6. Review Comments to the Author

Reviewer #5: I congratulate authors for the improved version of their manuscript.

There is still one area that must be looked at thoroughly, the discussion.

Authors have, in most of this section, repeated the results. Instead, they are required to provide an interpretation and implication of their findings. Furthermore, they should make sure that results are discussed by comparing them with similar studies. This has hardly been done.

7. PLOS authors have the option to publish the peer review history of their article (what does this mean?). If published, this will include your full peer review and any attached files.

**Do you want your identity to be public for this peer review?** For information about this choice, including consent withdrawal, please see our Privacy Policy.

Reviewer #5: No

---

## [Decision Letter · Decision Letter 4]

18 Apr 2024

PGPH-D-22-01896R4

Healthcare  delivery and keeping the workspace safe in the COVID pandemic: Results of a cross sectional survey of hospitals in India for the shaping of policies for future pandemics.

Dear Dr. Duggal,

Thank you for submitting your manuscript to PLOS Global Public Health. After careful consideration, we feel that it has merit but does not fully meet PLOS Global Public Health’s publication criteria as it currently stands. Therefore, we invite you to submit a revised version of the manuscript that addresses the points raised during the review process.

We look forward to receiving your revised manuscript.

Kind regards,

Miquel Vall-llosera Camps

Staff Editor

Journal Requirements:

Additional Editor Comments (if provided):

Reviewers' comments:

Reviewer's Responses to Questions

**Comments to the Author**

1. If the authors have adequately addressed your comments raised in a previous round of review and you feel that this manuscript is now acceptable for publication, you may indicate that here to bypass the “Comments to the Author” section, enter your conflict of interest statement in the “Confidential to Editor” section, and submit your "Accept" recommendation.

Reviewer #5: All comments have been addressed

2. Does this manuscript meet PLOS Global Public Health’s publication criteria? Is the manuscript technically sound, and do the data support the conclusions? The manuscript must describe methodologically and ethically rigorous research with conclusions that are appropriately drawn based on the data presented.

Reviewer #5: Yes

3. Has the statistical analysis been performed appropriately and rigorously?

Reviewer #5: Yes

4. Have the authors made all data underlying the findings in their manuscript fully available (please refer to the Data Availability Statement at the start of the manuscript PDF file)?

Reviewer #5: Yes

5. Is the manuscript presented in an intelligible fashion and written in standard English?

Reviewer #5: Yes

6. Review Comments to the Author

Reviewer #5: I wish to congratulate the authors for addressing on most of the comments. The manuscript is in better shape. However, there are still a few more issues that upon been addressed will improve the quality of this article significantly. They include the following:

1. Results section:

A. When assessing hospitals experiencing decreased hospitalization rates, the denominator should be the

hospitals that responded to that question and not all that were included in the survey.

B. Authors should consider summarizing the tables in their narrative by including only the most important

information.

2. Discussion:

A. Delete the table you have included in the discussion for it is enough to compare your findings, in the

paragraphs, with existing literature.

B. There is still in consistency in using the term COVID-19. It is capitalized in some places, missing the suffix 19

in others or in small cases.

C. Although the discussion has been improved, authors have not provided the interpretation or implication of

their findings or the similarities or differences observed with other studies. For instance where you have an

observation that large facilities were more likely to have separate facilities for COVID-19, would you suggest

to expand the sizes of small and medium-sized facility to provide comprehensive care in case of pandemic?

C2.Reduced essential supplies: information related to fears and insecurity are not relevant in this subject.

Instead include studies showing the consequences of reduced supplies during the pandemic and what

should be done.

C3. Health insurance and policies: Citing a study showing a similar or different result does not imply you

have discussed your findings. Rather, you are required to explain why you have that observation, and

why should it matter in a larger context. In this context, should the government and stakeholders, for

example, consider insuring more patients in pandemic situation?

7. PLOS authors have the option to publish the peer review history of their article (what does this mean?). If published, this will include your full peer review and any attached files.

**Do you want your identity to be public for this peer review?** For information about this choice, including consent withdrawal, please see our Privacy Policy.

Reviewer #5: No

---

## [Decision Letter · Decision Letter 5]

24 Jul 2024

PGPH-D-22-01896R5

Cautionary lessons from the COVID-19 Pandemic: Healthcare systems grappled with the dual responsibility of delivering COVID-19 and non-COVID-19 care.

Dear Dr. Duggal,

Thank you for submitting your manuscript to PLOS Global Public Health. After careful consideration, we feel that it has merit but does not fully meet PLOS Global Public Health’s publication criteria as it currently stands. Therefore, we invite you to submit a revised version of the manuscript that addresses the points raised during the review process.

We look forward to receiving your revised manuscript.

Kind regards,

Miquel Vall-llosera Camps

Staff Editor

Journal Requirements:

Additional Editor Comments (if provided):

Reviewers' comments:

Reviewer's Responses to Questions

**Comments to the Author**

1. If the authors have adequately addressed your comments raised in a previous round of review and you feel that this manuscript is now acceptable for publication, you may indicate that here to bypass the “Comments to the Author” section, enter your conflict of interest statement in the “Confidential to Editor” section, and submit your "Accept" recommendation.

Reviewer #5: All comments have been addressed

2. Does this manuscript meet PLOS Global Public Health’s publication criteria? Is the manuscript technically sound, and do the data support the conclusions? The manuscript must describe methodologically and ethically rigorous research with conclusions that are appropriately drawn based on the data presented.

Reviewer #5: Yes

3. Has the statistical analysis been performed appropriately and rigorously?

Reviewer #5: Yes

4. Have the authors made all data underlying the findings in their manuscript fully available (please refer to the Data Availability Statement at the start of the manuscript PDF file)?

Reviewer #5: Yes

5. Is the manuscript presented in an intelligible fashion and written in standard English?

Reviewer #5: Yes

6. Review Comments to the Author

Reviewer #5: I congratulate authors for addressing most of the comments.

Currently, the manuscript is technically and statistically better.

Very minor issues to address:

1. To be consistent on the use of the words COVID-19; there are some paragraphs which still contain the wrong word COVID without the suffix -19

2. While it is commendable to include as many comparative studies as possible in the discussion, it get a little bit annoying to add tables showing methods and results of these studies. I would suggest to retain texts and add information pertaining to the implications of their findings or discrepancies between the studies.

3. The following sentence from a paragraph in the methods "hospitals during May – July 202" has a wrong year. Please correct that

7. PLOS authors have the option to publish the peer review history of their article (what does this mean?). If published, this will include your full peer review and any attached files.

**Do you want your identity to be public for this peer review?** For information about this choice, including consent withdrawal, please see our Privacy Policy.

Reviewer #5: No

---

## [Editor Report · Decision Letter 6]

12 Sep 2024

Cautionary lessons from the COVID-19 Pandemic: Healthcare systems grappled with the dual responsibility of delivering COVID-19 and non-COVID-19 care.

PGPH-D-22-01896R6

Dear Professor Duggal,

We are pleased to inform you that your manuscript 'Cautionary lessons from the COVID-19 Pandemic: Healthcare systems grappled with the dual responsibility of delivering COVID-19 and non-COVID-19 care.' has been provisionally accepted for publication in PLOS Global Public Health.

Best regards,

Julia Robinson

Executive Editor